# Antispasmodic Activity of Light-Roasted Coffee Extract and Its Potential Use in Gastrointestinal Motility Disorders

**DOI:** 10.3390/foods13152307

**Published:** 2024-07-23

**Authors:** Acharaporn Duangjai, Anchalee Rawangkan, Atchariya Yosboonruang, Atcharaporn Ontawong, Surasak Saokaew, Bey-Hing Goh, Masami Suganuma, Pochamana Phisalprapa

**Affiliations:** 1Unit of Excellence in Research and Product Development of Coffee, Division of Physiology, School of Medical Sciences, University of Phayao, Phayao 56000, Thailand; atcharaporn.on@up.ac.th; 2Division of Microbiology and Parasitology, School of Medical Sciences, University of Phayao, Phayao 56000, Thailand or anchalee.ra@up.ac.th (A.R.); atchariya.yo@up.ac.th (A.Y.); 3Center of Health Outcomes Research and Therapeutic Safety (Cohorts), School of Pharmaceutical Sciences, University of Phayao, Phayao 56000, Thailand; surasak.sa@up.ac.th; 4Unit of Excellence on Clinical Outcomes Research and Integration (UNICORN), School of Pharmaceutical Sciences, University of Phayao, Phayao 56000, Thailand; 5Unit of Excellence on Herbal Medicine, School of Pharmaceutical Sciences, University of Phayao, Phayao 56000, Thailand; 6Division of Pharmacy Practice, Department of Pharmaceutical Care, School of Pharmaceutical Sciences, University of Phayao, Phayao 56000, Thailand; 7Biofunctional Molecule Exploratory Research Group (BMEX), School of Pharmacy, Monash University Malaysia, Bandar Sunway 47500, Malaysia; beyhingg@sunway.edu.my; 8College of Pharmaceutical Sciences, Zhejiang University, Hangzhou 310058, China; 9Sunway Biofunctional Molecules Discovery Centre, School of Medical and Life Sciences, Sunway University, Sunway City 47500, Malaysia; 10Graduate School of Science and Engineering, Saitama University, Saitama 3388570, Japan; masami0306@mail.saitama-u.ac.jp; 11Division of Ambulatory Medicine, Department of Medicine, Faculty of Medicine Siriraj Hospital, Mahidol University, Bangkok 10700, Thailand

**Keywords:** antispasmodic activity, light roast coffee extract, gastrointestinal motility disorders

## Abstract

Antispasmodic agents are crucial in managing gastrointestinal motility disorders by modulating muscle contractions and reducing symptoms like cramping and diarrhea. This study investigated the antispasmodic potential of different coffee bean extracts, including light coffee (LC), medium coffee (MC), and dark coffee (DC), on ileum contractions induced by potassium chloride (KCl), and elucidated their mechanisms of action using in vitro isolated tissue techniques. The results demonstrated that all coffee extracts reduced spontaneous contractions of rat ileum tissue in a dose-dependent manner. Among these, LC showed the most significant reduction in ileum contractions, particularly at higher concentrations. The key findings reveal that LC at 5 mg/mL significantly reduced CaCl_2_-induced contractions in isolated rat ileum tissue, indicating that LC may inhibit calcium influx or interfere with calcium signaling pathways. The presence of nifedipine, propranolol, and N-nitro-L-arginine methyl ester (L-NAME) have been confirmed in their involvement; they block calcium influx and calcium channels and activate β-adrenergic pathways as part of LC’s mechanism of action. The presence of their active compounds, particularly chlorogenic acid and caffeine, likely contributes to the observed antispasmodic effects. These findings suggest that LC exerts its antispasmodic effects by targeting key mechanisms involved in muscle spasms and intestinal motility, providing a potential for managing such conditions.

## 1. Introduction

Antispasmodic drugs are used to relieve muscle spasms and reduce intestinal motility, particularly in conditions such as abdominal cramping, diarrhea, and irritable bowel syndrome (IBS). These conditions significantly impact the health-related quality of life. Pharmacological treatments for these disorders include (i) anticholinergic agents (dicyclomine and hyoscyamine), which work by inhibiting the action of acetylcholine on muscarinic receptors; (ii) calcium channel blockers (pinaverium bromide and otilonium bromide) that block calcium channels in gastrointestinal smooth muscle; and (iii) an opioid agonist (trimebutine), which has agonist activity for the mu, kappa, and delta opioid receptors [1]. Recently, there has been a growing interest in traditional herbal alternatives for controlling motility disorders due to the low incidence of adverse drug reactions associated with natural treatments. Indeed, several essential oils, including L-menthol and peppermint oil, have been proposed as antispasmodic agents [2]. More specifically, the consumption of artichoke leaf extract can lead to an improvement in the quality of life for individuals suffering from IBS [3], a condition mediated by the effects of its active compounds, monoterpenoids, flavonoids, triterpenes, and alkaloids [4]. However, these extracts are applied peripherally to the skin, with poor absorption. Alternatively, coffee and related compounds, which are orally ingested, have been used as natural remedies [5].

The coffee plant (*Coffea arabica*) originates from Ethiopia and is classified under the Rubiaceae family. Coffee is a widely consumed beverage globally and is processed in several forms from roasted coffee beans. Numerous bioactive compounds contribute to its properties and potential health benefits. These include caffeine, chlorogenic acid, caffeic acid, trigonelline, diterpenes, and melanoidins [6]. Green and roasted coffee beans have been reported to be potential sources of active ingredients that exert favorable pharmacological activities, such as cardiovascular disease prevention, as well as antibacterial, antidiabetic, neuroprotective, and anticancer activities [6]. Evidence has been reported that caffeine relaxes smooth muscle by interfering with actin binding to phosphorylated myosin heads [7]. Caffeine also induces vascular smooth muscle relaxation by reducing calcium influx from intracellular calcium stores [8]. Moreover, chlorogenic acid stimulated mouse urinary bladder relaxation induced by carbachol, partly by enhancing cAMP levels by activating adenylyl cyclase [9].

The bioactive components of coffee, including chlorogenic acid and total phenolic compounds, as well as its antioxidant activity, are known to vary based on roasting-related factors such as heat level and roasting time. These components are generally reduced during the roasting process [10]. Despite the well-documented changes in these bioactive compounds, there is a notable lack of research on the antispasmodic effects of coffee bean extracts. Specifically, no study has yet demonstrated (i) the antispasmodic effect of coffee bean extract in vitro, (ii) the pharmacokinetic effects of different preparations, and (iii) the potential mechanisms of these effects. This study addresses this gap in the literature. The aims of this study were (i) to investigate the antispasmodic activity of coffee bean extract using both in vitro and animal models, (ii) to compare the differential activities of various coffee bean extracts, including unroasted coffee green bean (CGB) extract, light-roasted coffee (LC) extract, medium-roasted coffee (MC) extract, and dark-roasted coffee (DC) extract; furthermore, (iii) the study seeks to elucidate the possible mechanism(s) underlying the observed antispasmodic effects.

## 2. Materials and Methods

### 2.1. Chemicals

Krebs–Henseleit solution (10 mM HEPES, 122 mM NaCl, 5 mM KCl, 0.5 mM NaH_2_PO_4_, 0.5 mM KH_2_PO_4_, 1 mM MgCl_2_, 1.8 mM CaCl_2_, and 11 mM glucose), propranolol, nifedipine, and N-nitro-L-arginine methyl ester (L-NAME) were obtained from Sigma-Aldrich Corporation (St. Louis, MO, USA).

### 2.2. Plant Materials

Arabica coffee beans were provided by the Chao-Thai-Pukao Factory, Chiang Mai, Thailand. Voucher specimens were placed under voucher number NU003806 in the PNU herbarium of the Faculty of Biology at Naresuan University in Phitsanulok, Thailand. Coffee green beans (CGBs) were roasted to different degrees using different temperatures and times (10~20 min and 176.7~232.2 °C, respectively), and were categorized as light coffee (LC), medium coffee (MC), and dark coffee (DC) at the Chao-Thai-Pukao Factory by a coffee roaster (Silon ZR7, Coffee-Tech Engineering, Mazliah, Israel). Extraction was performed using ultrasound-assisted extraction (UAE) (Bandelin electric, Berlin, Germany). Coffee beans, CGB, LC, MC, and DC, were finely ground crude, then extracted with water (1:5; *w*/*v* sample to water) and sonicated at 35 kHz at 80 °C for 20 min. After sonication, an aqueous solution was filtered. The extraction process was repeated a third time. The mixture of the aqueous solution was dried in a ScanVac CoolSafe 110-4 Pro Freeze Dryer (Labogene A/S, Allerød, Denmark). Dried crude extracts of unroasted CGB, LC, MC, and DC were maintained at −20 °C until further study.

### 2.3. High-Performance Liquid Chromatography (HPLC) Analysis

Chromatographic analysis of coffee extracts was performed by separating the mobile phase A using 15% methanol; for mobile phase B, we used 85% methanol/DI water (30:70) with the addition of 2% acetic acid as needed to achieve a pH of 3.4. All mobile phases were filtered through a 0.45 μm filter before HPLC analysis. The LC extract was run with an Agilent 1200 series HPLC system (Agilent Technologies, Santa Clara, CA, USA), with the UV detector set at 320 nm for chlorogenic acid and ferulic acid, 280 nm for caffeine, and 327 nm for caffeic acid, and a Zorbax Eclipse XDB-C18 (4.6 × 150 mm, particle size 5 μm) (Agilent Technologies, Santa Clara, CA, USA) for 40 min using an injection volume of 20 µL with a flow rate of 0.6 mL/min. The chromatographic peaks were identified by matching their retention times to those of the standards.

### 2.4. Animals

Male Wistar rats weighing between 200 and 250 g were sourced from Nomura Siam International in Bangkok, Thailand. These rats were provided with a standard diet and water ad libitum and were housed in conditions maintained at 25 °C with appropriate humidity levels and a 12 h light/dark cycle at the Laboratory Animal Research Centre of the University of Phayao. Rats were acclimatized for one week before beginning the experiments. Each experimental group comprised 8–10 rats to ensure an adequate sample size for the study. Rats were fasted overnight before the experiments began and euthanized using exposure to CO_2_ under ethical guidelines. All procedures involving the animals were ethically reviewed and approved by the Animal Ethics Committee of the University of Phayao, Thailand, under approval number 620,104,009 to ensure compliance with animal welfare standards and regulations.

### 2.5. Experimental Protocol

The method used in this experiment was the same as that used and described in a previous study conducted by our research group [11]. Briefly, rats were fasted overnight and sacrificed. The ileum tissues were quickly isolated, and the mesenteric and fat attachments were removed. The tissues were then gently flushed and cleaned with cold Krebs–Henseleit solution. The ileum was cut into 1–1.5 cm sections. Each section was carefully placed in a 30 mL organ bath filled with Krebs–Henseleit solution at a temperature of 37 °C and supplied with a continuous flow of oxygen. The ileal sections were prepared under a consistent isotonic tension of 1 g and allowed to stabilize for 60 min before the tests. Isotonic responses measuring muscle contractions were monitored using a force transducer linked to an iWorx214 A/D converter and the LabScribe2 program (iWorx, Dover, NH, USA).

To evaluate the spasmolytic effect of the coffee extracts, 80 mM potassium chloride (KCl) was added to induce ileal contractions. After the start of contractions, each of the 4 coffee extracts was sequentially cumulatively added (0.01–10 mg/mL) to a different ileal section for each coffee extract. The observed minimal concentration of each of the four coffee extracts that affected the maximal relaxation response was used in further studies.

To understand the potential mechanism underlying the observed relaxation effect of LC on intestinal smooth muscle, 100 μM L-NAME, a nitric oxide synthase inhibitor, 1 μM propranolol, a β-adrenergic receptor antagonist, and 1 µM nifedipine, a calcium channel blocker, were individually added to the organ bath both with and without LC extract. CaCl_2_ (1–20 mM) was cumulatively added to an organ bath containing a Ca^2+^-free solution both with and without LC extract.

### 2.6. Statistical Analysis

The results represent the mean ± the standard error of the mean (SEM) from a minimum of six experiments per experimental group. One-way repeated measures analysis of variance (ANOVA) was employed to compare means across the four coffee extract groups, followed by Student’s *t*-test to assess statistical significance, with *p* < 0.05 considered significant. Statistical analyses were conducted using SPSS Statistics software for Windows (IBM Corp, Armonk, NY, USA).

## 3. Results

### 3.1. The Effect of Coffee Bean Extracts on Ileum Contraction Induced by KCl

We investigated the effect of different coffee bean extracts on ileum contraction induced by KCl. We sought to determine the impacts of CGB, LC, MC, and DC extracts on rat ileum tissue, specifically their potential as antispasmodic agents. Our results demonstrated that all tested coffee bean extracts decreased spontaneous contractions of the rat ileum in a manner that correlated with dosage (Figure 1 and Table 1). CGB extract demonstrated a gradual reduction in ileum contraction as the concentration increased, with the contraction percentage decreasing from around 95% at 0.01 mg/mL to approximately 88% at 10 mg/mL. LC showed a significant reduction in ileum contraction, particularly at higher concentrations. LC reduced ileum contraction at 5 and 10 mg/mL to around 80% and 75%, indicating a strong antispasmodic effect. MC exhibited a similar pattern to LC, with a steady decline in contraction percentage as the concentration increased. However, its effect was slightly less pronounced than that of LC. DC displayed the least reduction in ileum contraction among the extracts. Although there was a downward trend, the contraction percentage remained above 80% even at the highest concentration of 10 mg/mL. Among the extracts tested, LC shows the greatest potential to reduce ileum contractions, particularly at higher concentrations. No statistically significant difference was observed in the antispasmodic effect between LC at 5 mg/mL and LC at 10 mg/mL. Therefore, LC at 5 mg/mL was selected for further studies. These findings suggest that the roasting level and specific compounds present in the coffee extracts (such as chlorogenic acid) play a crucial role in their antispasmodic properties. Further investigation into their mechanisms is warranted to better understand these effects and their potential therapeutic applications.

### 3.2. The Antispasmodic Activity of LC Extract in the Presence of CaCl_2_ and Nifedipine

To explore the potential mechanisms underlying the antispasmodic activity of LC, cumulative doses of CaCl_2_ and nifedipine (a calcium channel blocker) were added to the organ bath with or without LC at 5 mg/mL. The impact of LC on the CaCl_2_-induced contractions in isolated rat ileum is depicted in Figure 2, where LC significantly attenuated the contraction response (*p* < 0.05). The reduced contraction responses with LC indicate that LC effectively inhibits ileum contractions induced by CaCl_2_. The reduction was significant at all tested concentrations of CaCl_2_ (*p* < 0.05), suggesting that LC has a robust antispasmodic effect. The data suggest that LC might be interfering with the calcium signaling pathways that mediate smooth muscle contraction. By inhibiting the calcium influx or the action of calcium within the smooth muscle cells, it could be a promising candidate for the treatment of gastrointestinal motility disorders. Further studies are needed to elucidate the precise mechanisms. Additionally, Figure 3 illustrates a significant difference in the spasmolytic effect of LC between conditions with and without nifedipine. High K^+^ induces ileum contractions by depolarizing the smooth muscle cells, leading to the opening of voltage-dependent calcium channels and an influx of calcium ions. Nifedipine, as a calcium channel blocker, significantly reduces K^+^-induced contractions, confirming the role of calcium influx in these contractions. LC also significantly reduces K^+^-induced contractions, indicating its potential as a calcium channel blocker or its ability to interfere with calcium signaling pathways. The combination of nifedipine and LC results in an even greater reduction in ileum contractions, suggesting that they may act synergistically or through different mechanisms to inhibit calcium-mediated contractions.

### 3.3. The Antispasmodic Activity of LC Extract in the Presence of Nitric Oxide Synthase [NOS] Inhibitor

To evaluate another possible mechanism of action, ileal tissue was incubated with LC either without or with an NOS inhibitor (Figure 4). High K⁺ alone induces ileum contraction, and was set as the 100% control. L-NAME, as an NOS inhibitor, significantly reduced the ileum contraction. LC also significantly reduced the ileum contraction, indicating that LC inhibits the contraction. However, there was no significant difference in the percentage of reduction in contractions between with and without L-NAME treatment. These findings suggest that the antispasmodic activity of LC does not involve the NO pathway.

### 3.4. The Antispasmodic Activity of LC Extract in the Presence of β-Adrenergic Antagonist

A third possible mechanism was examined. To evaluate the blockade of β-adrenergic pathways as a possible mechanism of antispasmodic activity, ileal smooth muscle was incubated with LC in the absence and presence of propranolol (β-adrenergic antagonist) (Figure 5). LC and propranolol significantly reduced the ileum contraction. There was a significantly greater reduction in contractility in ileal tissues incubated with LC alone than in those incubated with LC and propranolol (*p* < 0.01), indicating that they affect ileum contraction differently.

### 3.5. Determination of Active Compounds in Coffee Bean Extract by HPLC

Figure 6 shows a chromatogram of coffee bean extract analyzed by high-performance liquid chromatography (HPLC). The peaks in the chromatogram represent different compounds present in the coffee. The retention times shown in the chromatogram were as follows: chlorogenic acid—17.00 min; caffeic acid—20.00 min; ferulic acid—29.00 min; and caffeine—21.00 min. The heights and peak areas give an indication of the relative concentrations of active compounds in the extract. Chlorogenic acid and caffeine were present in significant amounts, as indicated by the larger peaks. The content of chlorogenic acid, caffeic acid, ferulic acid, and caffeine was found to be 24.80, 5.20, 0.44, and 39.90 mg/g extract, respectively.

## 4. Discussion

This research focused on evaluating the antispasmodic properties of coffee extract on intestinal smooth muscle to assess its potential application as a treatment for gastrointestinal motility disorders. The results indicated that coffee extracts had a relaxing effect on contractions of the rat ileum triggered by KCl. Of all the extracts and concentrations investigated, LC at 5 mg/mL demonstrated the greatest potential to reduce ileal contractions. To clarify the possible underlying mechanism(s), we investigated the action of LC in a calcium channel blockade or a reduced calcium influx and stimulation of nitric oxide production or β-adrenergic pathways.

Intestinal smooth muscle contractions are initiated by depolarization, which activates voltage-dependent Ca^2+^ channels and increases the intracellular calcium concentration due to a calcium influx and calcium release from the sarcoplasmic reticulum. Calcium binds with calmodulin, a calcium-binding protein, initiating a cascade of events that activate myosin light chain kinase (MLCK), resulting in the shortening of smooth muscle cells and the generation of force for contraction [12]. Relaxation of smooth muscle occurs as a result of reduced calcium influx or intracellular calcium or the blockade of calcium channels and excitatory or inhibitory neurotransmitters. The observed relaxation activity of LC was reduced in the presence of nifedipine, which suggests that the mechanism of LC is via calcium channels, calcium influx, and other mechanisms. A previous study reported that chlorogenic acid (CGA) suppressed Ca^2+^ influx from both the endoplasmic reticulum and outside the cells [13]. It has also been reported that caffeine induces vascular smooth muscle relaxation by inhibiting Ca^2+^ influx, and the authors suggested that caffeine blocks voltage-dependent Ca^2+^ channels, reducing calcium’s entry into the cytoplasm, and blocks the IP_3_ receptor [8,14]. This is consistent with another study, which found that caffeine suppressed calcium influx and inhibited calcium channels in myometrial smooth muscle cells [15]. Our findings suggest that the antispasmodic effect of LC may be due to the action of caffeine and chlorogenic acids. However, LC did not completely inhibit ileal contractions, suggesting the existence of other mechanisms.

Nitric oxide (NO) acts as a primary inhibitory neurotransmitter that plays a crucial role in controlling the tension of smooth muscles in various physiological processes including regulation of the digestive system [16]. NO interacts with soluble guanylate cyclase (sGC) to generate cyclic guanosine monophosphate (cGMP), leading to the relaxation of smooth muscles. The produced cGMP activates potassium (K^+^) channels, causing hyperpolarization of the cell membrane and subsequent muscle relaxation. Furthermore, cGMP inhibits the influx of calcium ions into the smooth muscle cells, which is crucial for maintaining relaxation and preventing excessive muscle contractions. In addition, cGMP-dependent protein kinase stimulates dephosphorylation of myosin light chains, leading to smooth muscle relaxation [17]. The antispasmodic activity of LC was not significantly different when used alone or in combination with L-NAME, indicating that LC’s mechanism of action does not rely on nitric oxide pathways. However, a previous study reported that chlorogenic acid induced endothelial vasodilation by increasing NOS [18]. Ferulic acid enhances the bioavailability of NO, a key signaling molecule involved in smooth muscle relaxation, thereby contributing to the vasodilation process [19]. Caffeine’s antiphosphodiesterase action leads to the accumulation of cAMP, which further raises non-contractile calcium levels, lowers intracellular calcium (iCa^2+^), and inhibits myosin light chain kinase (MLC Kinase), resulting in vasodilation [20]. Caffeine directly inhibits MLC Kinase and the interaction between actin and myosin [20], and caffeine indirectly affects vascular smooth muscle cells (VSMCs) through nitric oxide (NO). It acts on endothelial cells by increasing cytoplasmic Ca^2+^, which then forms a calcium–calmodulin complex. This complex activates the nitric oxide synthase enzyme, leading to the production of nitric oxide [21].

β-adrenoceptors are categorized into β_1_, β_2_, and β_3_ subtypes, and all these subtypes are present in smooth muscle tissue. In gastrointestinal tract smooth muscle, β_3_-adrenoceptors play a greater role in relaxation than β_1_-adrenoceptors. The stimulation of β-adrenoceptors by agonists increases intracellular cAMP levels via adenylyl cyclase activation. The elevation of cAMP induces cAMP-dependent protein kinase A (PKA) activation, which prevents the activation of the Ca^2+^–calmodulin complex via the phosphorylation of MLCK and increases Ca^2+^ reuptake into the sarcoplasmic reticulum, which leads to smooth muscle relaxation [22]. Current research has explored the impact of LC on ileal smooth muscle to understand its antispasmodic properties. This investigation involved studying the effects of LC in the presence and absence of propranolol, a β-adrenergic antagonist, to determine how LC interacts with β-adrenergic stimulation. Our results showed that the antispasmodic effects of LC decreased in the presence of propranolol, suggesting that LC acts via β-adrenergic pathways. However, LC may also exert its antispasmodic activity via one or more other mechanisms. Some studies have presented contradictory findings regarding the ability of coffee to induce relaxation in the proximal stomach and influence transit in the small bowel [23]. Interestingly, despite the observed relaxation effect on the proximal stomach, a randomized, controlled, crossover, single-blinded study using the barostat technique provided evidence that the study did not find any significant changes in gastric wall compliance or sensations following coffee consumption [24]. Another study found that coffee accelerated liquid-phase gastric emptying [25]. Additionally, the delayed gastric emptying time observed in newborn rats exposed to caffeine further underscores the impact of this compound on the movement of food through the stomach and into the intestines by involving ryanodine receptors [26]. In addition, caffeine-degrading compounds have a spasmolytic effect [27]. CGA has the ability to promote relaxation of the urinary bladder smooth muscle by enhancing the levels of cyclic adenosine monophosphate (cAMP) through the activation of adenylyl cyclase [9]. CGA has also been shown to elicit relaxant responses in the corpus cavernosum smooth muscle [28]. The possible mechanism of the antispasmodic action of LC may be influenced by CGA and caffeine. Understanding the molecular mechanisms underlying the relaxation of the ileal smooth muscle by the active compounds present in the coffee extract requires further investigation to elucidate the precise pathways involved in this process.

This study has some limitations. First, light, medium, and dark roast extracts were processed from beans roasted based on color. This subjective assessment is expected to vary between and among coffee roasters. Second, ileal contractility could vary among study rats. Third, information on the molecular mechanisms underlying the antispasmodic response of the active substances is limited.

## 5. Conclusions

This study indicated that coffee extract demonstrated antispasmodic properties. LC extract at 5 mg/mL exhibited the most significant antispasmodic effect. Our results suggest an antispasmodic mechanism of action of LC: it blocks calcium influx and/or calcium channels and activates the β-adrenergic pathways, which may be influenced by CGA and caffeine. These findings suggest that LC is a potential antispasmodic therapy for relieving muscle spasms and reducing intestinal motility. However, there are limitations and areas for future research. Further research involving clinical trials would provide a more comprehensive understanding of the efficacy and safety of these coffee extracts for managing gastrointestinal motility disorders. Future research could explore the long-term effects of coffee extracts on gastrointestinal motility and assess their potential for use in chronic conditions such as irritable bowel syndrome (IBS).

## Figures and Tables

**Figure 1 foods-13-02307-f001:**
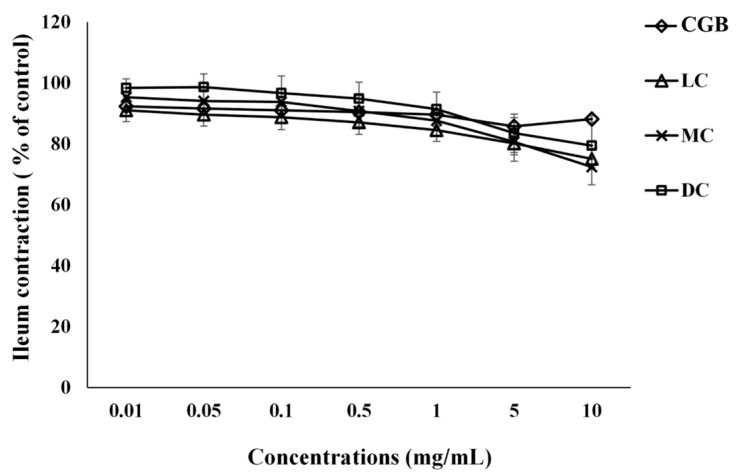
Spasmolytic effects of cumulative concentrations of coffee extract [coffee green bean extract (CGB), light roast coffee extract (LC), medium roast coffee extract (MC), and dark roast coffee extract (DC)] (0.01–10 mg/mL) on ileal contractions induced by 80 mM of KCl. Data are presented as mean ± SEM of 6–10 experiments.

**Figure 2 foods-13-02307-f002:**
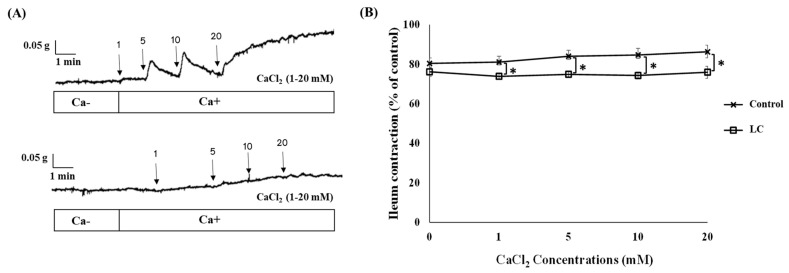
Cumulative concentration-response curves to CaCl_2_ both with and without the presence of light roast coffee extract (LC) (5 mg/mL) on ileal contractions. (**A**) Representative trace of LC extract and (**B**) ileum contractions with LC extract. Data are presented as mean ± SEM of eight experiments (* *p* < 0.05).

**Figure 3 foods-13-02307-f003:**
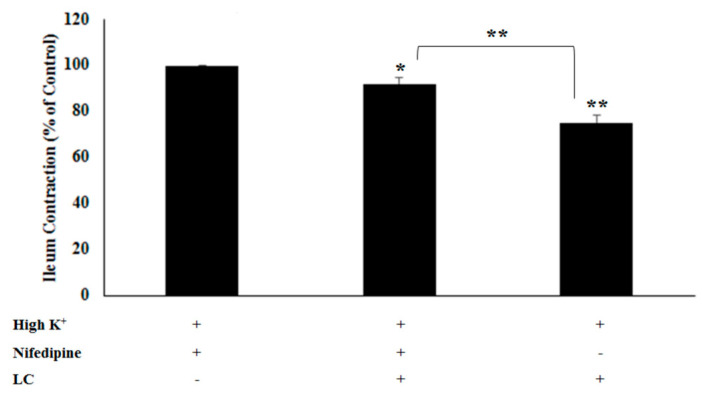
Effect of light roast coffee extract (LC) (5 mg/mL) on KCl-induced contractions of rat ileum was assessed both with and without 1 µM nifedipine (a calcium channel blocker). Results are presented as the mean ± SEM from eight experiments (* *p* < 0.05, ** *p* < 0.005).

**Figure 4 foods-13-02307-f004:**
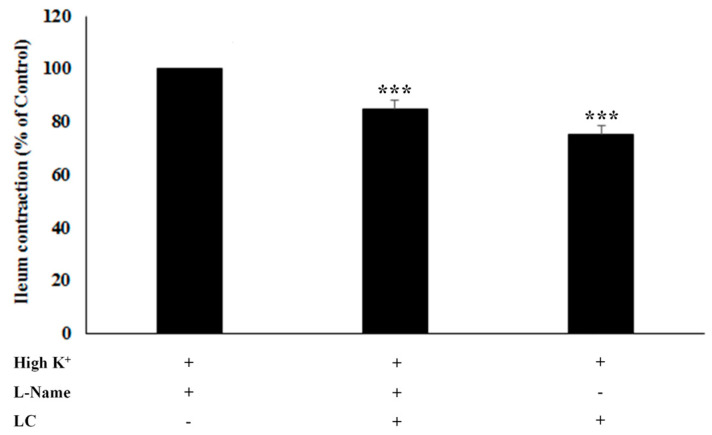
Effect of light roast coffee extract (LC) (5 mg/mL) on KCl-induced rat ileum contractions in the absence and presence of L-NAME (nitric oxide synthase inhibitor). Data are expressed as the mean ± standard error of the mean (SEM) of eight experiments (*** *p* < 0.001).

**Figure 5 foods-13-02307-f005:**
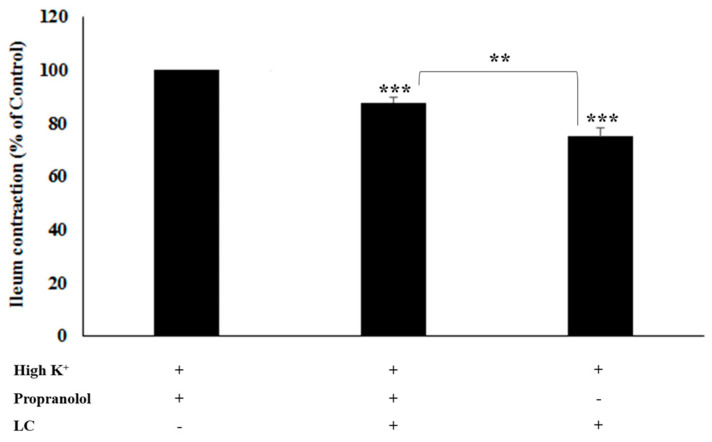
Effect of light roast coffee extract (LC) (5 mg/mL) on KCl-induced rat ileum contractions in the absence and presence of propranolol (β-adrenergic receptor antagonist). Data are expressed as the mean ± standard error of the mean (SEM) of eight experiments (** *p* < 0.01, *** *p* < 0.001).

**Figure 6 foods-13-02307-f006:**
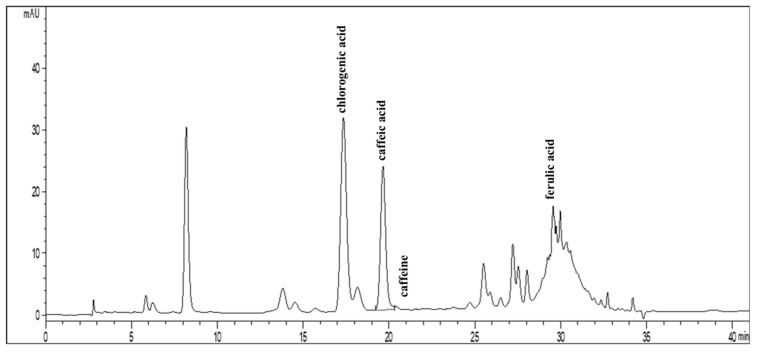
High-performance liquid chromatography (HPLC) chromatogram at 327 nm of light roast coffee extract (LC).

**Table 1 foods-13-02307-t001:** The effect of coffee extract on ileal contractions compared among the 4 evaluated coffee extracts.

Concentration (mg/mL)	Ileal Contractions (% of Control)			
	CGB Mean ± SEM	LC Mean ± SEM	MC Mean ± SEM	DC Mean ± SEM
**Control**	100 ± 0.0	100 ± 0.0	100 ± 0.0	100 ± 0.0
**0.01**	92.39 ± 1.25 **(*p* < 0.001)**	91.05 ± 3.64 **(*p* < 0.05)**	95.36 ± 3.01	98.40 ± 2.96
**0.05**	91.62 ± 1.42 **(*p* < 0.001)**	89.60 ± 3.72 **(*p* < 0.05)**	94.12 ± 3.81	98.66 ± 4.33
**0.1**	91.05 ± 1.42 **(*p* < 0.001)**	88.78 ± 4.05 **(*p* < 0.05)**	93.81 ± 4.24	96.70 ± 5.63
**0.5**	90.48 ± 1.41 **(*p* < 0.001)**	87.07 ± 3.88 **(*p* < 0.005)**	90.80 ± 5.06	94.91 ± 5.34
**1**	89.62 ± 1.34 **(*p* < 0.001)**	84.57 ± 3.68 **(*p* < 0.001)**	87.67 ± 5.26 **(*p* < 0.05)**	91.43 ± 5.59 **(*p* < 0.05)**
**5**	85.76 ± 2.90 **(*p* < 0.001)**	80.18 ± 3.73 **(*p* < 0.001)**	80.73 ± 6.35 **(*p* < 0.05)**	83.55 ± 6.30 **(*p* < 0.005)**
**10**	88.20 ± 1.07 **(*p* < 0.001)**	75.12 ± 3.42 **(*p* < 0.001)**	72.43 ± 5.81 **(*p* < 0.001)**	79.50 ± 7.19 **(*p* < 0.005)**

A *p*-value < 0.05 indicates statistical significance compared to control. SEM, standard error of the mean; CGB, coffee green bean extract; LC, light roast coffee extract; MC, medium roast coffee extract; DC, dark roast coffee extract.

## Data Availability

The original contributions presented in the study are included in the article material, further inquiries can be directed to the corresponding authors.

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
