# Peer review of "Antispasmodic Activity of Light-Roasted Coffee Extract and Its Potential Use in Gastrointestinal Motility Disorders"

_foods, 2024, doi:10.3390/foods13152307_

Round 1

Reviewer 1 Report

Comments and Suggestions for Authors

Reviewer comments:

The authors have presented their research work in a manuscript entitled “Antispasmodic activity of light-roasted coffee extracts and its potential use in gastrointestinal motility disorders.” The work done is good, and the manuscript is written well. The manuscript can be accepted for publication in the ‘Foods’ journal after addressing some minor revisions.
The following are the comments:

1.      Line 31: Write ‘investigated’.

2.      Line 155: Start this section using a background statement.

3.      What are the shortcomings/bottlenecks with respect to implementing the findings of the present study using the antispasmodic effect of coffee extract with anti-spasmodic therapy for relieving muscle spasms and reducing intestinal motility in the industry? What else needs to be done in this work to actually apply this method in the medical industry?

4.      The results of the present study have not been properly compared with the reported studies from the literature. Also, I suggest adding results and discussion together to improve the quality of the manuscript.

5.      The conclusion section can be improved with the addition of major results and values.

Author Response

We would like to thank the editor and reviewers for their careful and thorough review of this manuscript. We have revised our manuscript in response to your suggestions. We hope that this improved manuscript is acceptable for publication.

The revisions in the manuscript are highlighted, and the answers to your specific comments and suggestions are as an attachment file. 

Reviewer 2 Report

Comments and Suggestions for Authors

my comments are in the attached pdf

Comments on the Quality of English Language

minor

Author Response

We would like to thank the editor and reviewers for their careful and thorough review of this manuscript. We have revised our manuscript in response to your suggestions. We hope that this improved manuscript is acceptable for publication.

The revisions in the manuscript are highlighted, and the answers to your specific comments and suggestions are as follows:
